# Should you become a leader in online collaborative learning? Impact of assigned leadership on learning behaviors, outcomes, and perceptions

**Heng Luo**[1]*, **Xu Han**[1], **Ying Chen**[1], **Yanjiao Nie**[2]

**1** Faculty of Artificial Intelligence in Education, Central China Normal University, Wuhan, China, **2** Affiliated Elementary School of Hubei University, Wuhan, China

* luoheng@mail.ccnu.edu.cn

**Data Availability Statement:** The research data are available from the Mendeley Data database, and can be accessed openly at https://data.mendeley.com/datasets/797wsfgppy/1.

## Abstract

The growing prevalence of collaborative learning spaces in higher education highlights the importance of student leadership for group learning. Thus, leadership assignment as a common practice in online collaborative learning merits special attention. To investigate the impact of assigned leadership and its key characteristics in promoting team learning in the online context, a semester-long quasi-experiment was conducted with 94 students in a graduate-level blended course. The results revealed significant differences between assigned leaders and group members in certain participating behaviors. However, the impact of assigned leadership on learning outcomes and perceptions was insubstantial. Additionally, student leaders' academic achievement was found to have little impact on group members' learning behaviors and learning outcomes, and mixed results were reported regarding the influence of leader behaviors on group performance. The research findings can inform the crucial decision of leader selection and extend our understanding of leadership in online collaborative learning.

## Introduction

Research on leadership is prevalent in the domains of organization and management science, where chief executive officers (CEOs) and managers are commonly investigated leadership roles [1–3]. More recently, researchers have shifted their attention to the context of computer-supported collaborative learning (CSCL), where students are divided into groups for collaborative knowledge construction and problem solving [4, 5]. In particular, the importance of leadership has been emphasized in the context of online collaborative learning, due to its great demand on learner autonomy and social interactions [5–7].

The literature has investigated student leadership in online collaborative learning from various perspectives, including effective leadership behaviors [8, 9], distinctive leadership styles [10], moderation effect of learning task [11, 12], leadership emergence and distribution [7, 13, 14], and effectiveness of peer-led group learning [15, 16].

**Funding:** The authors received no specific funding for this work.

**Competing interests:** The authors have declared that no competing interests exist.

While prior findings have deepened our understanding of online leadership, most studies examined its impact from a holistic perspective, focusing on group-level performance and experience rather than individual differences between leaders and members within groups. Consequently, an ethical question remains unanswered: Does the leadership role bring unique benefits to student leaders and make them advantageous over group members in online collaborative learning? Additionally, since high-achieving students are more likely to assume leadership roles in collaborative learning [17], this question of equity becomes more relevant, prompting us to investigate the impact of leadership assignment on online learning behaviours, outcomes, and perceptions in the present study.

## Review of relevant literature

### Definition and classification of leadership

Despite being important for collaborative knowledge building and problem solving [18, 19], leadership lacks a universally accepted definition in literature. There are three main perspectives on the nature of leadership: a social status within a group with featured responsibilities [20], a combination of capabilities to satisfy a group's needs and achieve shared goals [10], and a dynamic social process to solve problems through social interaction and resources management [21]. In this study, we refer to leadership as a social status, but critically examine the emergence and impact of such a social status from the perspectives of capabilities and social processes.

Based on how leadership status is obtained or emerged in the process of collaboration, we classified leadership into three types. The first type is assigned leadership. As its name suggests, leader status is granted from top to bottom. In general, it is believed that official appointments can motivate leaders to perform their leading duties (e.g., facilitate collaboration) more actively, resulting in better team performance [16, 22]. Yet, Bacon et al. [23] cautioned that random assignment of leadership is highly subjective to chance, and thus, the effects of assigned leadership cannot be guaranteed. Due to its operational convenience, assigned leadership is most commonly practiced in the management process [24, 25], and is more frequently investigated in the literature [10, 11].

The second type is emerged leadership. Unlike assigned leadership, this type is formulated in a bottom-up fashion and emerges spontaneously without official assignment. There are two preconditions for its occurrence: 1) the absence of an assigned leader or the negligence of the assigned leader, and 2) the presence of group members with strong working and organizational capabilities [13, 26]. It is found that members with high intelligence, active participation, and positive self-views are more likely to emerge as leaders [7, 13, 27].

The third type of leadership is distributed leadership, in which the duties of the leader are shared among several members within the group [28–30]. In this scenario, several members volunteer to undertake partial duties based on their specific skill sets [27, 28]. Consequently, the boundary of distributed leadership is vaguer and more inclusive, allowing the benefits of leadership status to be shared among more people [31, 32].

### Leadership in online collaborative learning

Most research studies on leadership in online collaborative learning focus on how leadership promotes group performance. Koeslag-Kreunen et al. [11] conducted a meta-analysis with 43 empirical studies and concluded that leadership is an essential factor to facilitate team learning in terms of engagement and performance, regardless of the leadership source and style. Exemplary performance and positive comments from leaders have been found to promote group morale and confidence [33], supporting the close association between collective self-efficacy

and effective modelling [34]. Additionally, competent leaders can promote peer interaction and a harmonious learning atmosphere, two essential conditions for group cohesion, which in turn leads to improved academic performance and learning satisfaction [6, 35].

Compared to group learning performance, leadership studies on individual performance within groups are limited. A few researchers investigated the influence of leadership status on student leaders themselves, particularly their learning performance in terms of behavioral, cognitive, and emotional engagement. Research findings show that leader role was associated with enhanced behavioral engagement, indicated by more frequent login and posting behaviors [22]. Moreover, student leaders were found to be more cognitively active than other members in online discussions [16]. Lastly, student leaders also demonstrated higher emotional engagement featured by a sense of ownership, enhanced motivation, and greater self-efficacy [17, 22, 36].

Compared to student leaders, few studies focus exclusively on the individual performance of group members. Since group performance is usually leader-driven [10, 11], problematic member performance often gets hidden under the strong leader performance. Scanty evidence was reported on member reaction to leadership in online collaborative learning, including collective development of higher-order thinking [37–39] and reduced sense of insecurity [40]. According to Leader-Member Exchange (LMX) theory, relationship between the leaders and members is mutual, and member traits such as extraversion and agreeableness are important predictors of leadership effectiveness and LMX quality [41, 42].

## Research gaps and research questions

While there has been a growing body of research on student leadership in online collaborative learning, several gaps exist in the literature that undermine the credibility and interpretability of the research findings. First, the complexity of leadership construct has not been properly addressed. Various theoretical classifications of leadership exist because of the distinction in its conceptual nature, acquisition mechanism, and object of influence. However, such distinction has been inadequately analyzed in empirical studies, resulting in oversimplified research findings that lack theoretical sophistication.

Second, the influences of leadership on leaders and members in online collaborative learning have often been investigated separately, without correlational analysis of their interplay. For example, Burke et al. [10] limited their investigation of essential behaviors and learning outcomes to student leaders only, while Rourke and Anderson [39] explored the collective benefits of leadership (e.g., group cohesion and engagement) without role-specific comparative analysis.

Lastly, in assessing the treatment effects of leadership in online collaborative learning, many studies lacked a sufficient duration of investigation. For example, Choi et al. [33] admitted that the findings based on 5-day temporary training groups were not persuasive enough. Yilmaz et al. [43] further argued that poor group cohesion caused by insufficient study duration can coexist with leadership effects. Furthermore, time constraints also limit the number and complexity of online learning tasks, leading to reduced generalizability of research findings [12, 44].

To address the above limitations, this study investigated the impact of assigned leadership on the learning performance and experience between assigned leaders and group members during a 12-week blended course. The primary purpose of the present study is to uncover the unique learning benefits associated with the assigned leadership role and explore how those benefits change over time and with varying learning tasks. Additionally, this study seeks to

identify the key characteristics of assigned leadership to promote group performance. More specifically, the following questions guided our research inquiry:

1. What is the impact of assigned leadership on student performance in online collaborative learning in terms of behaviors, outcomes, and perceptions? What are the influencing factors?

2. What are the key characteristics of assigned leadership that can effectively promote group learning performance in online collaborative learning?

## Methods

### Ethics statement

The research study was conducted in accordance with the ethical standards of the Helsinki Declaration. The research procedures and instruments were reviewed and approved by the Institutional Review Board of Central China Normal University (CCNU-IRB-201909021, approved on 2019/09/16). Written informed consent forms were obtained from all participants before the study. All participants were made aware that their participation in the study was voluntary, and their personal identifiable information would be kept anonymous at all publications and presentations. Participants can withdraw from the research study anytime, without penalty.

### Participants and research context

A total of 94 graduate students from a research university in central China participated in this quasi-experimental study during the 2019 fall semester. The participant group comprised 86 women and 8 men. They were all first-year students admitted to a graduate program of educational technology, with an average age of 23.6 (ranging from 21 to 25). The study was implemented in a blended course called instructional design and case analysis, which lasted for 12 academic weeks. Students were required to apply teaching and learning theories to solve authentic instructional problems through group activities such as case analysis, online discussion, and collective reflection. This course was selected for investigating the research topic due to its complexity and ill-structured nature, which are known to promote collaborative learning and group interaction [4, 14]. All participants took the blended course for the first time, with no prior blended or online learning experiences.

In the first class, the participants were randomly assigned into 24 groups (22 groups of four and 2 groups of three). In each group, a group leader role was randomly assigned to one student using Excel's RAND function. Consequently, there were a total of 24 assigned group leaders (1 man and 23 women) and 70 group members. After random assignment, student leaders were made aware of their three main leadership responsibilities: (1) informing and reminding group members of the learning tasks, (2) encouraging group members to participate in online discussion, and (3) coordinating group assignment completion. The leadership responsibilities were recommended but not mandatory, and would not affect student leaders' final course grade.

The blended course comprised 12 face-to-face lecture sessions and weekly online discussions. The lecture sessions were organized to cover key topics of instructional design (ID) in the following sequence: *front-end analysis* (3 weeks), *design and development* (3 weeks), *implementation and evaluation* (3 weeks), *course review* (2 weeks), and *final exam* (1 week). After each lecture session, students participated in three types of CSCL activities in a Moodle-based discussion forum (https://www.wolearn.org/). These included (1) reading discussions, where

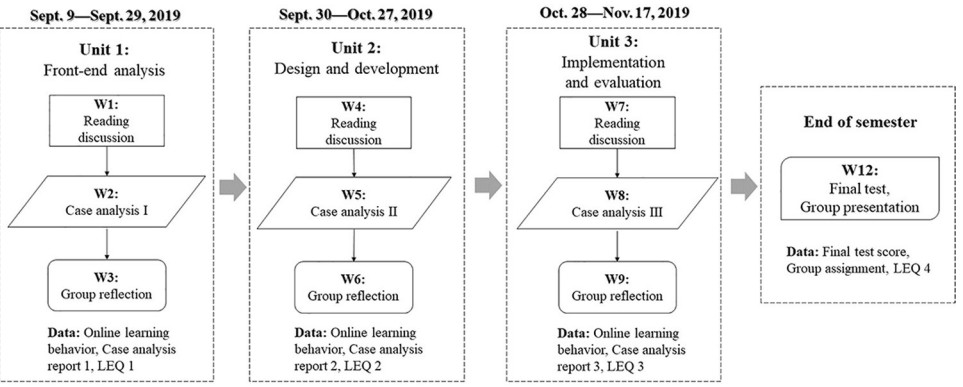

**Fig 1. The blended learning activities and overall research process.**

students shared their notes and thoughts from assigned reading materials (e.g., textbook chapter, journal articles), (2) case analysis, in which students worked in groups to analyze and solve a given ID case problem, and (3) group reflection, where students jointly reflected on their learning experience in the current instructional unit. A sub-forum was created for each weekly activity, but students could continue to post in the sub-forum when the activity was past due. In addition to group discussion forums, we also included a whole-class forum to allow both within-group and between-group discussions.

## Procedure

The overall research procedure is illustrated in Fig 1. The online collaborative activities occurred primarily in the first 9 weeks of the course and were grouped into three instructional units based on the topic. Each unit consisted of three types of weekly tasks: reading discussion, case analysis, and group reflection. At the end of each instructional unit, each student group was required to submit a case analysis report indicating the key issues of and the best solution to the given case problem. Each student was also required to submit a learning experience questionnaire (LEQ) to rate their perceived online learning experience during the instructional unit. Additionally, students' learning behaviors in the discussion forum were dynamically captured in the forum logfiles and databases. In the last week, students individually participated in a 90-minute paper-based closed exam and submitted the final group assignment as assessments of individual learning and group learning outcomes. The LEQ was administered for the fourth time in Week 12 to collect students' overall perception of online collaborative learning during the course.

## Data collection methods

Three types of quantitative data were collected in this study to measure students' learning behaviors, learning outcomes, and learning perception. The variables and their operations, data sources, and collecting instruments are listed in Table 1.

Learning behaviors were measured by the number of course login, forum views, posts created, posts replied, and total words of all posted content, which were key indicators of behavioral engagement. The behavioral data were captured automatically by the course platform in its logfiles and database forms.

Learning outcomes were measured by the grading scores of the final exam and final group assignment. Both instruments were developed by course instructors with their validity verified through three cycles of course iterations (2016–2017). The final exam comprised two parts: 15

**Table 1. Types of data collected in this study and the variables they measured.**

| Construct | Variable | Operation | Source | Instrument |
|---|---|---|---|---|
| **Learning behaviours** | *login* | Number of logins to the course | Discussion forum | Log files |
| | *forum_view* | Total views of discussion posts | Discussion forum | Log files |
| | *post* | Number of posts created | Discussion forum | Log files |
| | *reply* | Number of replies to other students | Discussion forum | Log files |
| | *total_words* | Number of words in all posts | Discussion forum | Database |
| **Learning outcomes** | *test_score* | Score of the final exam | Grading scores | Final exam |
| | *obj_score* | Score of objective items | | |
| | *subj_score* | Score of subjective items | | |
| | *report_ rating* | Quality of the final group assignment | Grading scores | Group Assignment |
| **Learning perception** | *LEQ_rating* | Average rating of all questionnaire items | Student rating | LEQ |

LEQ, learning experience questionnaire.

objective test items accounting for 40 points and two subjective items (i.e., analysis and application), accounting for 60 points. The final group assignment required students to identify an instructional problem and provide an ID report as a group with a detailed analysis and solution proposal. The grading of the subjective test items and ID report was based on four assessment criteria suggested by Reigeluth and Frick [45]: relevance, comprehensiveness, depth, and theoretical underpinning. To ensure the accuracy of the grading results, two researchers rated the subjective test items and the ID report independently after reaching a good inter-rater reliability (Spearman's Rho > 0.9) through training. The mean scores of the two raters were used as the final grading scores. It is important to note that the test scores measure individual learning outcomes, while the report rating, based on the group assignment, measures the group learning performance.

Learning perception was measured by the mean score of LEQ, which comprised nine five-point Likert scale items regarding students' collaborative learning experience (see S1 Appendix) and was administered four times during the research process. The Cronbach's α values computed from the four LEQ datasets were 0.839, 0.881, 0.864, and 0.936, respectively, indicating good internal reliability.

## Data analysis methods

One-way ANOVA was used to analyze the grading scores and behavioral data as they fit the assumptions of normality and homogeneity of variance. A non-parametric test (Mann-Whitney U) was conducted with the LEQ ratings, which are ordinal data in nature. Furthermore, a correlation analysis was employed to explore the relationships between leadership characteristics and group performance in online collaborative learning. Because the normality assumption was satisfied for most behavior and outcome variables, we selected Pearson's r as the correlational coefficient.

Additionally, social network analysis was conducted to examine the patterns of social interaction, as it is considered "an appropriate method for revealing relational structures that arise from CSCL interactions." [46]. We are particularly interested in network density as a global measure to describe group cohesion during online collaborative learning. A denser network indicates a higher level of group participation and collaboration [22].

We also performed a qualitative analysis of selected student leaders and members to enable triangulation and meaningful interpretation of the statistical results. Six student groups were purposefully selected as cases of interest due to their typicality or idiosyncrasy of group

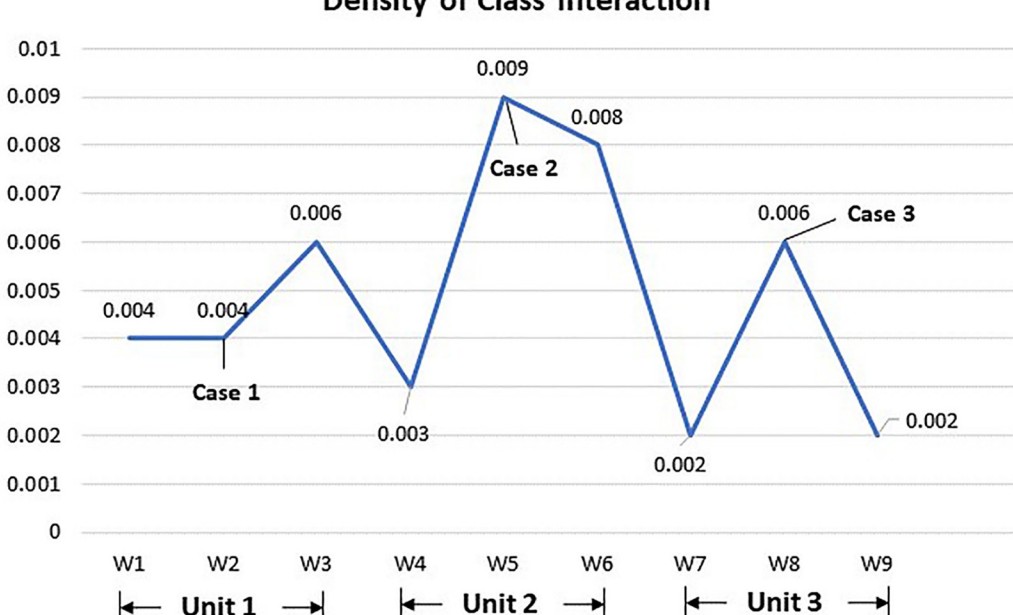

**Fig 2. Density of class interaction in the discussion forum.**

dynamics and performance. They were subjected to further qualitative analysis using the coding methods described by Saldaña [47] (e.g., process coding, emotion coding, and taxonomic coding). In particular, the sequential interactions within the discussion groups were visually demonstrated, and the text contents of all posts were thematically analyzed and classified. The qualitative results provide justification and elucidation of the quantitative findings and reveal hidden trends and patterns regarding various types of leadership in CSCL.

## Results

### Overall social learning pattern

The variation in network density revealed several interesting findings regarding the overall social learning patterns in the blended course. As shown in Fig 2, the density of class interaction in the discussion forum fluctuated drastically throughout the course, with three peaks emerging in Weeks 3, 5, and 8. Unlike Unit 1, which reported the highest density during the weekly activity of group reflection, the other two instructional units both witnessed a surge of density during the weekly activity of case analysis. Moreover, there appears to be a reverse U-shaped relationship between density and time. The overall density of class interaction during the middle phase of the semester was higher than that in the early and later phases. It should also be noted that the overall density remained at a low level ($\rho < 0.01$) despite changes over time, indicating relatively poor group cohesion and online collaboration.

### Difference in learning behavior

We compared five types of learning behaviors between assigned leaders and group members during nine weeks of online learning and plotted the means in Fig 3. In general, student leaders surpassed group members in the behaviors of course login, forum viewing, and posting. This suggests the effectiveness of assigned leadership in promoting certain learning behaviors.

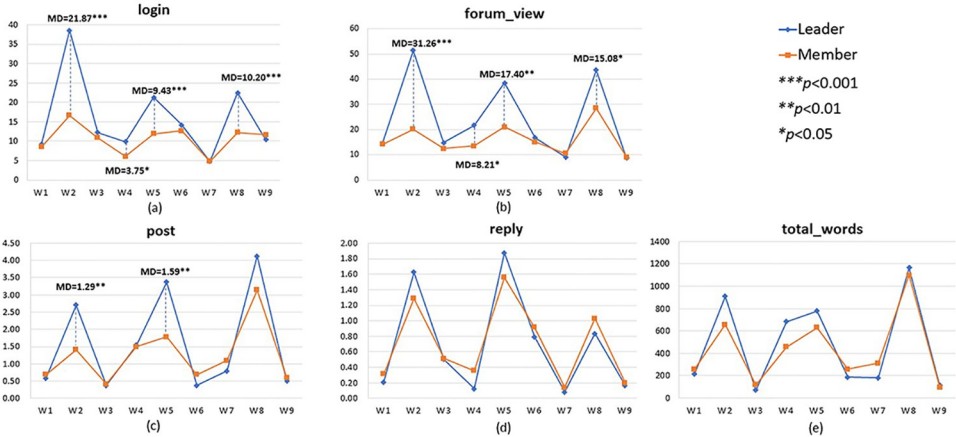

**Fig 3. Participating behaviors of leaders and members.**

Moreover, the type of learning tasks seems to influence the variance in learning behaviors, as significant behavioral differences were only identified in Weeks 2, 5, and 8, where students engaged in the weekly activity of case analysis. Interestingly, no major differences were found in the behaviors of replying and total words, as assigned leaders and group members demonstrated similar behavioral patterns throughout the course.

## Difference in learning outcome

Fig 4 displays three types of mean scores, obtained by assigned leaders and group members in the final exam, as their learning outcomes: objective test scores, subjective test scores, and total test scores. In general, the difference in learning outcome seemed to be insubstantial. Despite the 2.13 margin, one-way ANOVA results revealed no significant difference in students' total test scores and group members even obtained slightly higher scores in the subjective test. The only statistically significant mean difference (MD) was identified with the objective test scores (MD = 2.28, $F = 4.035$, $p = 0.048 < 0.05$), suggesting that the assigned student leaders may have a slight advantage over group members in online learning outcomes.

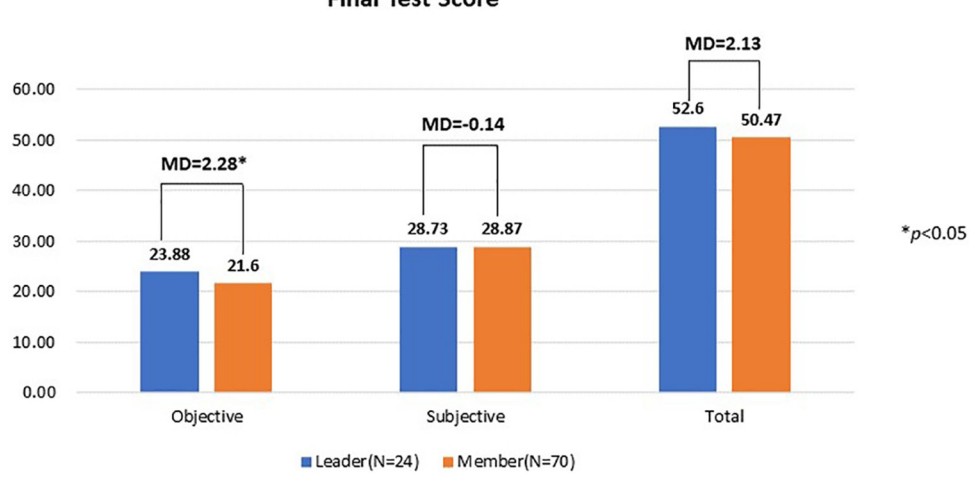

**Fig 4. Mean difference in three types of test scores between leaders and members.**

## Difference in learning perception

The mean LEQ ratings throughout the semester for assigned leaders and group members are shown in Fig 5. All ratings ranged between 4 and 4.5, on a five-point Likert scale and demonstrated a gentle ascending pattern. This suggests a positive and consistently improving learning perception of online collaborative learning for all students. Furthermore, the Mann-Whitney U test revealed no significant difference in the mean LEQ ratings between leaders and members in the four surveys, indicating that assigned leadership does not necessarily lead to enhanced learning perception. Despite the overall similitude of learning perception, further analysis has identified only one item that received significantly higher ratings from student leaders: *No.4*: *I have promoted collaboration within the group*. This item measures students' sense of ownership in the online collaborative learning.

## Leader characteristics and group performance

To identify the essential leader characteristics that effectively predict member performance in online collaborative learning, we calculated the Pearson's r coefficients between assigned leaders and group members in terms of learning behaviors (i.e., login, forum view, post, reply, total words) and learning outcomes (i.e., test score and report rating). The results are presented in Table 2. In general, leaders' behavioral engagement had little impact on members' learning behaviors, with the action of replying being the only exception. Leaders' replying behavior was found to strongly correlate with that of the members (r = 0.6), and moderately affected the members' forum viewing and posting behaviors (r = 0.49 and 0.47). Interestingly, the leader behaviors of login and forum viewing even adversely affected member performance in the final exam (r = −0.56 and −0.58). Moreover, the academic achievement of assigned leaders appeared to be an irrelevant predictor of member performance in CSCL, as the final test scores of the student leaders were found to be unrelated to the learning behaviors and test scores of the group members. Nonetheless, the influence of student leaders on collaborative

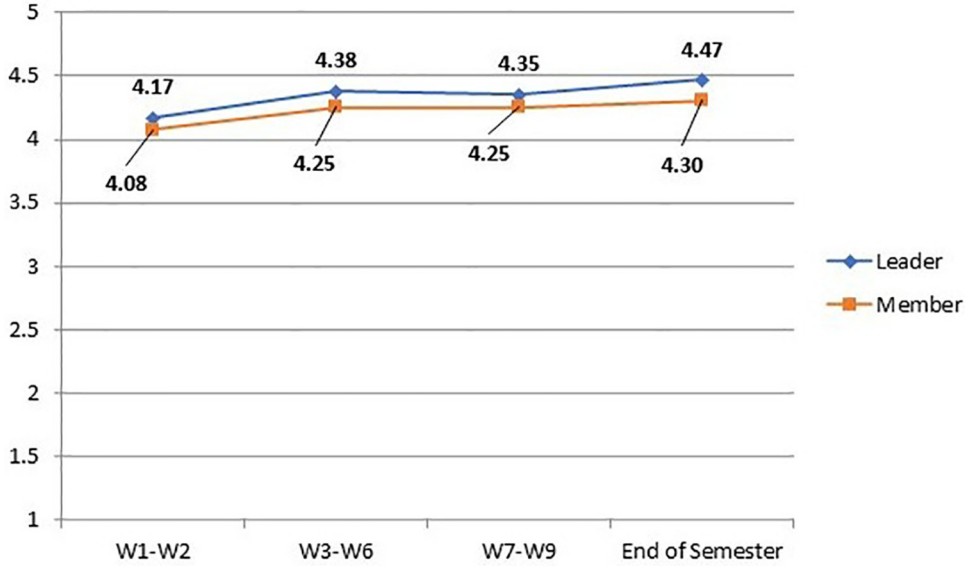

**Fig 5. Mean LEQ ratings for leaders and members during the course.**

**Table 2. Correlations between leader performance and member performance.**

| Member / Leader | Login | Forum view | Post | Reply | Total words | Test score (individual) | Report rating (group) |
|---|---|---|---|---|---|---|---|
| **Login** | −0.01 | 0.09 | 0.21 | 0.16 | −0.01 | *−0.56*** | *0.56*** |
| **Forum view** | 0.03 | 0.21 | 0.27 | 0.28 | 0.13 | *−0.58*** | **0.65*** |
| **Post** | 0.04 | 0.33 | 0.24 | 0.34 | 0.20 | −0.3 | *0.52*** |
| **Reply** | 0.02 | *0.49** | *0.47** | **0.60*** | 0.40 | −0.21 | *0.51** |
| **Total words** | 0.01 | 0.40 | 0.39 | 0.34 | 0.39 | −0.30 | **0.60*** |
| **Test score** | −0.06 | 0.05 | 0.10 | −0.04 | −0.03 | −0.14 | **0.63*** |

Bold text indicates a strong correlation ($r \geq 0.6$), italic text indicates a medium correlation ($0.6 > r \geq 0.4$); ***$p < 0.001$

**$p < 0.01$

*$p < 0.05$.

performance should not be ignored, as their learning behaviors and learning outcomes effectively predict the quality ratings of the group assignment (for all paired variables, $r > 0.5$).

## Emerged and distributed leadership

In this study, the leader status was randomly assigned regardless of students' academic achievement, self-efficacy, and willingness to lead. A closer look at the social interactions within each group revealed two additional types of leadership: emerged leadership and distributed leadership. In these two types of leadership, the functional identity of a group leader was not appointed, but rather acquired through commanding presence and contributing behaviors over time.

Fig 6 illustrates two examples of emerged leadership identified in Group N (five threads) and Group K (one thread). In Fig 6, the posts created by the instructor and the students are represented in oval and rectangular shapes, and the linking arrows show the number and sources of the replies received. The dates of posting and coded instructional functions are assigned to each post to indicate the sequence and nature of peer interactions.

As shown in Fig 6, student N10 is considered an emerged leader who demonstrated leading behaviors more frequently than the assigned leader (N21) during the weekly discussion. In addition to answering the required questions posted by the instructor, N10 took initiative to post additional questions for discussion and encouraged other students to participate in humorous and supporting comments. For example, when no one responded to a question posted by herself, N10 jokingly remarked, "I guess I will just have to answer my own question then." When someone answered the question, N10 promptly offered her gratitude and compliments. In contrast, the assigned leader, N21, participated passively with only three replies. Further, we noticed that high-achieving students were more likely to acquire leader status during online collaboration through substantial cognitive contribution. A typical example is student K40, who contributed enormously to a thread of discussion by providing reflective questions, quality insights, critical evaluation, and supplementary resources.

The two examples shown in Fig 6 were rare cases in online collaborative learning where assigned leaders failed to fulfil their responsibilities. More commonly, we noticed the phenomenon of distributed leadership featured by the concurrent presence of both assigned and emerged leaders in one group. Assigned leaders demonstrated more coordinating behaviors such as posting questions, sending out reminders, and synthesizing viewpoints, while emerged leaders engaged with more socio-cognitive activities such as replying, sharing, emotional recognition, and evaluation. Additionally, we noticed that distributed leadership tended to occur in groups with multiple high-achieving students.

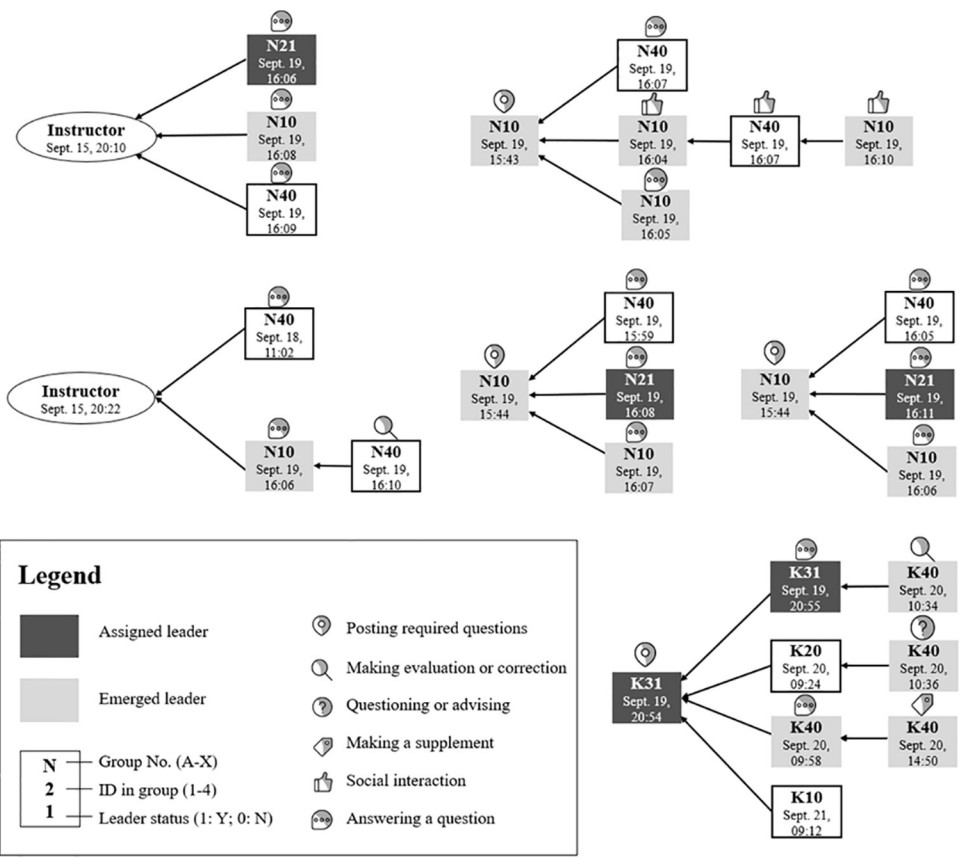

**Fig 6. Visualization chart showing assigned and emerged leadership based on sample.**

## Discussion and perspectives

In contrast to earlier findings regarding the various advantages associated with leadership status [16, 22, 36], this study indicates only conditional benefits of assigned leadership in increasing participating behaviors such as login, viewing, and posting during the weekly task of case analysis. The fact that the reading and reflection tasks witnessed no significant behavioral increase among assigned leaders suggests the moderation effect of learning tasks on leadership impact, with performance-oriented and open-ended tasks associated with increased leader engagement. This finding supports Mercier et al. [12]'s argument that "if students were engaging in a task that was less collaborative or less cognitively demanding, there may have been less need for the amount of leadership. . .".

Contrary to the common belief that leadership positively influences academic performance [5, 17], our study revealed no role-based differences in students' online learning outcomes as measured by the final test scores. Further analysis indicated that assigned leaders slightly outperformed group members in the objective test, which was designed to assess students' knowledge, recall, and comprehension. This finding corroborates Zha and Ottendorfer [16]'s discovery of student leaders' better achievement in lower-order rather than higher-order thinking. According to Zha and Ottendorfer [16], topic familiarity and time restraint might hinder student leaders from initiating and engaging in online discussion at a higher-order cognitive level.

The random assignment of the leadership role in this study might explain the lack of difference in online learning perception between leaders and members. The literature indicates a reinforcing cycle between leader attributes and leader role. Students with desirable personal traits such as social self-efficacy, intelligence, conscientiousness, and extraversion are more likely to emerge or become nominated as group leaders [8, 17, 27]. The same traits are also known to enhance learning perception and satisfaction [48, 49]. Through randomized appointment, we severed the association between leader role and leader attributes in this study, and the results indicate that leader role does not guarantee improved learning perception, as the added responsibility might negatively affect student leaders' learning experiences with extra workload, increased anxiety, and social conflicts.

Regarding the essential leader characteristics, we found the strong influence of leader behaviors and academic achievement on group assignment quality particularly worrisome. The excessive reliance on leaders' effort and contribution in group works implies the prevalence of "free-riders" or "bystanders" in online collaboration [50, 51]. The fact that leaders' participating behaviors (login and forum view especially) negatively correlate with members' learning performance confirms our speculation that a dedicated leader might be perceived, by others, as a "babysitter." Thus, the likelihood of free-riding and over-dependence increases, leading to poor individual learning outcomes for group members during online learning. Lastly, replying behavior proved to be the only leader behavior that predicted member participation. According to Kim et al. [7], replying with individualized messages is considered a person-focused leadership behavior, which induces positive emotions among group members, and more frequent member presence and engagement in online collaboration as a result.

Consistent with the existing literature [10, 27, 52], instances of emerged and distributed leadership were also reported in this study, and two types of relationship between assigned and non-assigned leadership roles were identified. The first type is substitution, where an assigned leader failed to fulfill leading responsibilities and was replaced by an emerged leader during the collaborative learning process. The phenomenon of substitution is expectable with random assignment of leader role, where unwilling or incapable students have equal chance to get elected as group leaders. The second type is distribution, where the emergence of on-assigned leadership was more contingent with the attributes of a particular group member (e.g., academic achievement, personality, etc.) rather than the involvement of the assigned leader. Consequently, distributed leadership should not be viewed as a social status, but rather a preferred relationship between assigned and emerged leadership.

We believe that the existence of non-assigned leadership is confounded by the impact of assigned leadership in this study, as the benefits of leadership were transferred to the emergent leaders or distributed within the groups. Similarly, the insubstantial influence of assigned leaders on group learning performance should not be interpreted as the insignificance of the leader role in online collaborative learning. Rather, the variance in assigned leaders' behavioral engagement and academic achievement might be offset by the representation of emergent leadership. After all, it is the individual presence and behaviors, rather than the assigned status, that matters the most in team effectiveness and productivity [10].

## Implications

Our research findings have several implications. For students, it is generally beneficial to become a group leader in CSCL, as it brings increased behavioral engagement and slightly better lower-order learning outcomes. For teachers, when selecting student leaders, social responsibility matters more than academic achievement in promoting group engagement. Thus, low-achieving students deserve equal opportunities to lead in CSCL. For student leaders, the

preferrable leadership style should be person-focused and facilitative; too much direct involvement and personal contribution might adversely affect individual member learning. For group members, one's presence and behaviors in CSCL should not be bounded by the assigned leader identity since emerged and distributed leadership often brings additional benefits such as enhanced social interaction and cognitive contribution.

## Limitations and future research

Three limitations should be noted when interpreting research findings. First, despite our best efforts, extraneous factors that threatened internal validity were unable to be eliminated. For instance, students' engagement and group cohesion in CSCL were inevitably affected by the events and progress of the academic semester. Additionally, the awareness of leader identity is likely to cause demoralization or compensatory rivalry among group members, obscuring the causal effect of leadership. Second, the participants were predominantly women from a single course, which undermines the generalizability of the study results to other CSCL contexts as the results might be gender-biased and driven by course-specific knowledge. Lastly, the empirical data collected in this study lacked diversity for more accurate measurements and meaningful interpretations. For example, CSCL engagement was measured using only five learning behaviors, and interview data with student leaders and group members were non-existent. Consequently, we recommend future research to investigate the impact of assigned leadership in varied CSCL contexts (e.g., co-ed groups from various disciplinary domains) with more rigorous design and diversified empirical data.

## Supporting information

**S1 Appendix. Learning experience questionnaire items.**
(DOCX)

## Acknowledgments

The authors acknowledge the invaluable contributions of Liyuan Wei, Min Zhu, Peiyu Wang, Jiaxin Yang, and Siyi Jiang from Central China Normal University for their assistance in data collection and analysis. The authors also want to thank the participated students for their support in this study.

## Author Contributions

**Conceptualization:** Heng Luo.

**Formal analysis:** Xu Han, Ying Chen, Yanjiao Nie.

**Investigation:** Yanjiao Nie.

**Methodology:** Heng Luo.

**Supervision:** Heng Luo.

**Visualization:** Xu Han.

**Writing – original draft:** Heng Luo, Xu Han.

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
