## [Decision Letter · Decision Letter 0]

30 Dec 2021

PONE-D-21-35819Should you become a leader in online collaborative learning? Impact of assigned leadership on learning behaviors, outcomes, and perceptionsPLOS ONE

Dear Dr. Luo,

Thank you for submitting your manuscript to PLOS ONE. After careful consideration, we feel that it has merit but does not fully meet PLOS ONE’s publication criteria as it currently stands. Therefore, we invite you to submit a revised version of the manuscript that addresses the points raised during the review process.

We look forward to receiving your revised manuscript.

Kind regards,

Rong Zhu, Ph.D.

Academic Editor

PLOS ONE

2. Please change "female” or "male" to "woman” or "man" as appropriate, when used as a noun (see for instance https://apastyle.apa.org/style-grammar-guidelines/bias-free-language/gender).

*In order to improve reporting, in your methods section, please provide additional information about the demographic details of your participants.

Reviewers' comments:

Reviewer's Responses to Questions

**Comments to the Author**

1. Is the manuscript technically sound, and do the data support the conclusions?

Reviewer #1: Partly

2. Has the statistical analysis been performed appropriately and rigorously? 

Reviewer #1: N/A

3. Have the authors made all data underlying the findings in their manuscript fully available?

Reviewer #1: Yes

4. Is the manuscript presented in an intelligible fashion and written in standard English?

Reviewer #1: No

5. Review Comments to the Author

Reviewer #1: Below are comments for the authors to consider.

Comments:

1. This study tests the effect of assigned leadership on online behaviors, outcomes, and perceptions, using data of 94 student-course observations from a China's university. The author needs to explain why the course is suitable for investigating this topic.

2. The students who more skewed to the female is could be one of the limitations. I suggest the author use more student-course observations to overcome gender bias.

3. The results may be driven by the course-specific knowledge. I suggest the author should employ more comprehensive data including all courses at the university for each student.

4. It wold be useful to further investigate and discuss the relationship between assigned leaders, emerged leaders, and distributed leaders.

5. I would also encourage you to engage a professional copyeditor to polish the writing of the paper.

6. PLOS authors have the option to publish the peer review history of their article (what does this mean?). If published, this will include your full peer review and any attached files.

Reviewer #1: No

---

## [Author Response · Author response to Decision Letter 0]

8 Feb 2022

We greatly appreciate this opportunity to revise our manuscript and address the potential issues identified by the editor and the reviewer. We are very grateful for their careful review and insightful feedback. Our point-by-point responses to the individual comments made by each reviewer are listed in our response letter. Please also refer to the tracked changes for detailed revisions.

---

## [Editor Report · Decision Letter 1]

25 Mar 2022

Should you become a leader in online collaborative learning? Impact of assigned leadership on learning behaviors, outcomes, and perceptions

PONE-D-21-35819R1

Dear Dr. Luo,

We’re pleased to inform you that your manuscript has been judged scientifically suitable for publication and will be formally accepted for publication once it meets all outstanding technical requirements.

Kind regards,

Rong Zhu, Ph.D.

Academic Editor

PLOS ONE

---

## [Editor Report · Acceptance letter]

29 Mar 2022

PONE-D-21-35819R1 

Should you become a leader in online collaborative learning? Impact of assigned leadership on learning behaviors, outcomes, and perceptions 

Dear Dr. Luo:

I'm pleased to inform you that your manuscript has been deemed suitable for publication in PLOS ONE. Congratulations! Your manuscript is now with our production department. 

Kind regards, 

on behalf of

Dr. Rong Zhu 

Academic Editor

PLOS ONE